# REDNOTE-VIBE: A DATASET FOR CAPTURING TEMPORAL DYNAMICS OF AI-GENERATED TEXT IN SOCIAL MEDIA

## ABSTRACT

The proliferation of Large Language Models (LLMs) has led to widespread AI-Generated Text (AIGT) on social media platforms, creating unique challenges where content dynamics are driven by user engagement and evolve over time. However, existing datasets mainly depict static AIGT detection. In this work, we introduce RedNote-Vibe, the first longitudinal (5-years) dataset for social media AIGT analysis. This dataset is sourced from Xiaohongshu platform, containing user engagement metrics (e.g., likes, comments) and timestamps spanning from the pre-LLM period to July 2025, which enables research into the temporal dynamics and user interaction patterns of AIGT. Furthermore, to detect AIGT in the context of social media, we propose PsychoLinguistic AIGT Detection Framework (PLAD), an interpretable approach that leverages psycholinguistic features. Our experiments show that PLAD achieves superior detection performance and provides insights into the signatures distinguishing human and AI-generated content. More importantly, it reveals the complex relationship between these linguistic features and social media engagement. The code and dataset will be publicly available.

## 1 INTRODUCTION

Large Language Models (LLMs) (Achiam et al., 2023; Mallick & Kilpatrick, 2025; Guo et al., 2025) have revolutionized digital content creation, leading to a surge in AI-Generated Text (AIGT). Consequently, developing robust detection methods has become a critical research frontier, with a focus on classification (Gui et al., 2025; Hu et al., 2023) and source attribution (Sun et al., 2025). However, existing research primarily treats AIGT detection as a static classification task on formal corpora (e.g., news, academic writing), a paradigm that misaligns with the unique ecosystem of social media.

In this work, we identify two critical and unaddressed challenges for AIGT research on social media. First, unlike formal text, where factual accuracy is prioritized, **social media ecosystem rewards content that maximizes user engagement**, such as likes, comments, and shares (Chung et al., 2023; Cascio Rizzo et al., 2024). On social media platforms that value sharing real-life experiences, LLMs can be prompted to generate sensationalized or controversial content to increase interactions, which undermines connections and trust within the community. Second, **over time, AI and human content are interacting more frequently on social media platforms.** This creates a potential co-evolution of linguistic styles, which may also influence community topics, user behavior, and engagement patterns.

Existing datasets and research paradigms typically treat AIGT detection as a static classification task on a fixed snapshot of data, fail to capture these longitudinal trends or explain the relationship between linguistic features and engagement metrics. This research gap is not merely a technical oversight but an approaching cultural issue, as the social media ecosystem could be transforming rapidly. Based on this, measuring and identifying these dynamics is of paramount importance.

To bridge this gap, we introduce RedNote-Vibe, the first dataset designed for studying AIGT in a dynamic social media context. RedNote-Vibe is collected from Xiaohongshu (RedNote), a leading Chinese social media platform. Each sample is enriched with metadata including topic, tags, timestamp, user engagement metrics (i.e., likes, comments, collections) and their parallel AIGT variants generated by a diverse set of LLMs. Notably, our data collection spans a wide timeline, covering content from before the release of ChatGPT to the present (July 2025). This provides a natural testbed for researchers to observe the evolution and impact of LLMs within social media environments.

To address the limitations of existing detection models, which lack the necessary interpretability to link linguistic artifacts to user engagement, we propose the PsychoLinguistic AIGT Detection Framework (PLAD). PLAD first quantifies text into a suite of psycholinguistic features and then utilizes a decision tree-based model for classification. Our experiments show that PLAD not only achieves superior performance with model-based methods but also offers clear insights into the stylistic signatures of different LLMs and reveals how AIGT correlates with user engagement. The main contributions of this paper are as follows:

- We introduce RedNote-Vibe, a social media AIGT dataset featuring rich engagement metadata and a longitudinal timeline, enabling research on temporal dynamics and user interaction patterns.

- We propose PLAD, an interpretable, psycholinguistic-based framework that offers strong detection performance while illuminating the connection between linguistic style and social media engagement.

- We provide comprehensive analysis of our dataset, uncovering temporal trends in AI adoption and revealing differences in engagement patterns between human-authored and AI-generated content.

## 2 RedNote Engagement Dataset

RedNote (Xiaohongshu)[1] stands as one of the most influential Chinese social media platforms, serving over 300 million monthly active users. This platform emphasizes personal experiences and lifestyle sharing, which makes it particularly vulnerable to AIGC infiltration, as it undermines the authenticity of the content. Despite its influence, RedNote has remained largely unexplored in academic research due to the absence of a publicly available dataset. In this section, we present our RedNote-Vibe dataset, the first large-scale dataset from this platform that captures both temporal dynamics and engagement patterns, specifically designed for research on AIGT detection and the impact of AIGC on social media.

### 2.1 Data Collection and Statistics

Our data collection methodology is grounded in RedNote's official user behavior report, which identifies ten dominant content categories: *Career, Wellness, Travel, Health, Food, Pets, Education, Sports, Fashion, and Relationships*. We first extract the example tags provided in the report for each category, then expand them to approximately 50 representative tags per topic through manual curation. These expanded tag sets serve as our retrieval queries to ensure comprehensive coverage of each domain.

We adopt a web crawler to collect 120,000 notes from January 2020 to July 2025. To ensure accurate topic classification, we filter these notes using Qwen-2.5-7B, resulting in 98,714 notes. Each sample contains comprehensive metadata including: 1) Content: note title, text content and tags; 2) Temporal information: publication timestamp; 3) Engagement metrics: likes, comments and collections; 4) Topic domain.

Table 1 presents detailed statistics across domains, which shows that different domains exhibit distinct linguistic properties (e.g., average length) and engagement dynamics. Figure 1 visualizes the distribution of the total engagement (defined as the sum of all three metrics), which follows a long-tail pattern consistent with real-world social media. As shown in the inner ring of the chart, a substantial proportion of posts receive very low engagement (0-10 interactions), indicating that most user-generated content attracts limited audience attention. In contrast, a small fraction of posts achieve disproportionately high engagement.

[1]https://www.xiaohongshu.com

Table 1: Statistics of the RedNote posts across content categories. The post count (#) is presented in thousands (k). Comm. and Colls. refer to the average number of comments and collections, respectively.

| Domain | # | Length | Likes | Comm. | Colls. |
|--------|------|--------|-------|-------|--------|
| Health | 14.9k | 398.0 | 383.3 | 38.8 | 322.0 |
| Fashion | 11.4k | 273.0 | 718.6 | 40.9 | 265.9 |
| Food | 7.9k | 427.0 | 49.0 | 11.2 | 35.5 |
| Career | 11.1k | 485.8 | 381.5 | 72.6 | 294.6 |
| Pets | 2.1k | 457.0 | 33.9 | 11.0 | 18.9 |
| Education | 4.8k | 529.8 | 47.0 | 5.9 | 33.5 |
| Sports | 4.4k | 559.6 | 38.6 | 9.4 | 21.3 |
| Relation. | 15.5k | 372.5 | 522.2 | 171.9 | 183.8 |
| Travel | 10.5k | 549.0 | 373.4 | 59.8 | 295.4 |
| Wellness | 10.7k | 483.6 | 552.5 | 110.9 | 282.1 |

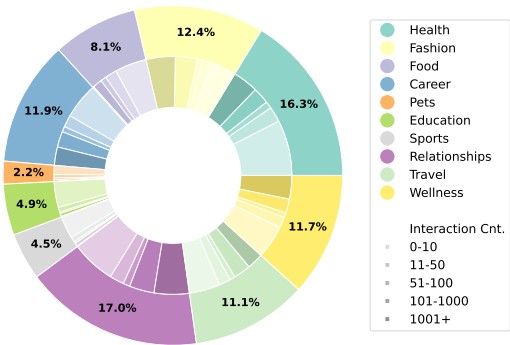

Figure 1: The distribution of posts by domain and interaction count (sum of likes, comments and collections).

## 2.2 AI Samples Generation

To create a comprehensive AIGT detection benchmark, we construct parallel AI-generated versions of RedNote posts. Our generation protocol employs a seed-based approach: each human-written note serves as a reference for LLMs to generate stylistically similar but semantically distinct content. Specifically, we provide LLMs with the original note's title, content, and domain classification, then instruct them to create new posts that: (1) emulate the writing style and tone of the reference; (2) preserve personal characteristics such as colloquialisms, punctuation patterns, and occasional grammatical imperfections typical of social media; (3) maintain comparable text length; and (4) avoid direct copying of phrases or sentences from the original post.

We create a model pool comprising 17 representative LLMs from 6 providers. All models receive identical prompts with JSON-formatted output. Notably, we exclusively select seed notes published before November 2022 (pre-LLM period) to ensure their human-authored property. These seeds are randomly distributed across the LLM pool, with each model generating at least 1,000 samples.

Figure 2 illustrates our data construction pipeline and the selected LLM ensemble. This approach yields a training and validation set with verified human/AI labels. Additionally, we compile an exploration set containing posts from the post-LLM period (2023-2025). While lacking ground-truth labels, this subset enables researchers to investigate real-world content evolution and analyze the emerging linguistic landscape shaped by widespread AI adoption.

## 2.3 Task Definition

Leveraging our dataset's rich structure, we define three hierarchical classification tasks that reflect real-world AIGT detection scenarios with increasing granularity:

- **AIGT Classification (binary)**: Human vs. AI-generated text detection task, which requires models to distinguish human-written content from any AI-generated text. This task establishes the baseline capability for AIGT detection.

- **AI Provider Identification (6-way)**: A task focusing on the AI-generated subset, where models identify the source among six major AI providers (OpenAI, Google, Anthropic, etc.). This task tests whether detection methods can capture family-level patterns, as models from the same provider often share similar training methodologies and corpus.

- **Model Identification (17-way)**: A fine-grained AI model identification task, where we distinguish between 17 specific AI models within the AI-generated content, representing the most challenging scenario that requires detecting subtle model-specific characteristics.

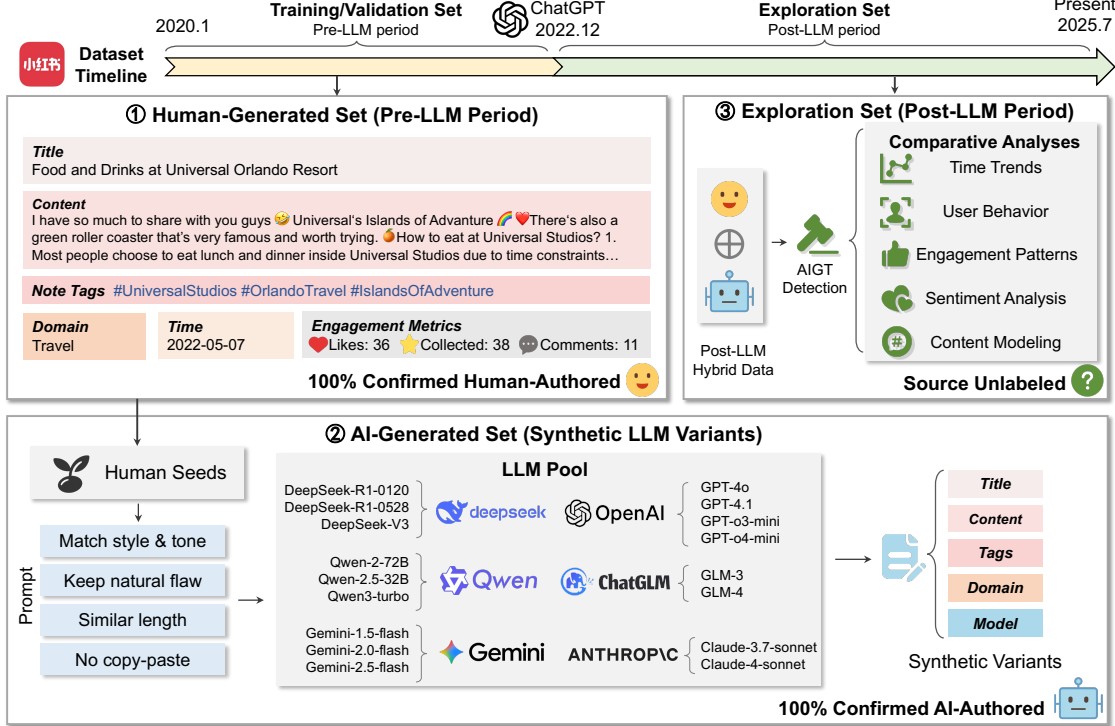

Figure 2: Overview of the RedNote-Vibe dataset construction. **(1) Human-Generated Set** is collected in the pre-LLM period before 2022.12, therefore labeled as human-authored. **(2) AI-Generated Set** is created by prompting a diverse LLM pool with the human seeds to produce synthetic variants. **(3) Exploration Set** contains post-LLM social posts, enabling AIGT detection and extensive temporal and cross-sectional analyses comparing AI and human content.

## 3 PSYCHOLINGUISTIC AIGT DETECTION FRAMEWORK

To address the unique challenges of AIGT detection in dynamic social media environments, we propose the PsychoLinguistic AIGT Detection Framework (PLAD). Unlike existing methods that often fail to explain why a text is classified as AI-generated, PLAD leverages established psychological theories to create an interpretable detection system that not only distinguishes AI from human content but also reveals the underlying linguistic mechanisms.

### 3.1 PSYCHOLINGUISTIC FEATURE FRAMEWORK

Our framework quantifies a total of 31 linguistic features around four dimensions of human language expression, each rooted in psychological and cognitive theories. The definition of dimensions is shown below:

**Emotional and Social Grounding.** Human communication is deeply rooted in personal experience and social awareness. Authentic emotional expression is often linked to memory and is conveyed through rich sensory details Conway & Pleydell-Pearce (2000). Furthermore, humans naturally adapt their language to their audience, demonstrating an implicit Theory of Mind through markers of empathy and social connection (Baron-Cohen, 1997). We capture these aspects through features measuring emotional intensity, personal grounding, social connection, and specific markers such as emoji usage patterns (Felbo et al., 2017).

**Cognitive Architecture.** Drawing from conceptual complexity theory (Baker-Brown et al., 1992) and narrative structure research (Labov & Waletzky, 1997), human authors exhibit multiperspectival reasoning and tolerance for ambiguity. Human argumentation characteristically incorporates nuanced counterarguments

(Toulmin, 2003) and embeds coherent value systems (Graham et al., 2009). This dimension encompasses features that quantify perspectival complexity, dialectical reasoning, and temporal coherence.

**Lexical Identity and Stylistic Signature.** Over time, human writers develop a unique stylistic idiolect (Argamon et al., 2003), characterized by word choices, rhythmic patterns, and natural imperfections such as hesitations, self-corrections, and topic shifts (Clark & Tree, 2002), which signal real-time cognitive processing. While LLMs can imitate various styles, their output exhibits a stochastically uniform distribution of words that reveals their non-human origin. This dimension focuses on quantifying the uniqueness of the lexicon, the stylistic consistency, and the presence of natural linguistic imperfections that signal the human author.

**Cohesion and Textual Flow.** While the previous dimension assesses word choices, this dimension evaluates the organization and progression of the full content. The human composition naturally evolves through interconnected ideas, exhibiting dynamic semantic progression and adaptive referential chains (Halliday & Hasan, 2014). In contrast, AI-generated text often maintains paragraph-level fluency while lacking deep thematic development, producing semantically static content. We also measure repetition patterns: humans employ strategic repetition for emphasis or clarification (Stamatatos, 2009), which is rarely seen in AI text.

### 3.2 FEATURE EXTRACTION AND CLASSIFICATION

Our feature set contains 31 features that can be categorized into two extraction approaches. (1) Directly computable statistical features that can be obtained using straightforward computational methods, such as emoji density, type-token ratio and other structural measures. (2) Semantically-based features that require evaluation criteria and text analysis tools for assessment. In contrast to traditional approaches to psychological text analysis such as LIWC (Pennebaker et al., 2015), which employs frequency-based word analysis, we adopt a more sophisticated approach using a proxy LLM for psychological text analysis.

Specifically, following related work (Rathje et al., 2024; Ghatora et al., 2024), we design evaluation rubrics that convert theoretical constructs into measurable criteria for characteristics. These rubrics are then presented to the proxy LLM, who is instructed to evaluate the input text according to the specified dimensions and provide quantitative scores. To mitigate potential biases introduced by proxy models, all results undergo a verification mechanism using Chain-of-Thought reasoning to ensure accuracy. Experimental validation demonstrates that our configuration achieves higher correlation with human annotations compared to existing methods. The detailed feature list and example of criteria are shown in the Appendix.

Based on the extracted feature vector $\mathbf{f}(x) \in \mathbb{R}^{31}$, we train a supervised classifier to predict the text's label. To ensure the framework's interpretability, we utilize tree-based models such as XGBoost and CatBoost, which provide clear feature importance rankings. The classification task is formally defined as finding the label $\hat{y} = \arg\max_{y \in \mathcal{Y}} P(y|\mathbf{f}(x))$, where $\mathcal{Y}$ is the set of possible labels. The model is trained by minimizing the cross-entropy loss, $\mathcal{L}_{\text{CE}} = -\sum_i y_i \log(p_i)$.

## 4 EXPERIMENTS

### 4.1 AIGT DETECTION

**Experiment Setup.** To evaluate the performance of our proposed PLAD, we compare it with two categories of approaches. (1) **Classical statistics-based methods**: StyloAI (Opara, 2024), Binoculars (Hans et al., 2024) and the method of Ullah et al. (2024), which serve as representative feature-driven baselines for AIGT detection. (2) **Model-based methods**: This category covers strong baselines, including (i) fine-tuning pre-trained text classification models such as BERT-base (Devlin et al., 2019), RoBERTa-base (Liu et al., 2019), and ALBERT-base (Lan et al., 2019), which represent state-of-the-art paradigms (Gritsai et al., 2024; Li et al., 2024), and (ii) established AIGT detection pipelines such as Sniffer (Li et al., 2023), POGER (Shi et al., 2024), and LLM-Idiosyncrasies (Sun et al., 2025). We follow the original training protocols of all methods and fine-tune them on our dataset. For comparison, we evaluate our proposed PLAD framework with different classifiers, including CatBoost (Prokhorenkova et al., 2018), XGBoost (Chen & Guestrin, 2016), and Gradient Boosting Classifier (Friedman, 2001).

Table 2: Comparison of PLAD with existing methods across detection tasks derived from RedNote-Vibe. All results are shown as percentages without the % symbol. Acc. denote accuracy. Precision, recall, and accuracy are computed as macro-averages.

| Method | Model Identification (17-way) | | | Provider Identification (6-way) | | | AIGT Classification (binary) | | |
|--------|-----------|--------|------|-----------|--------|------|-----------|--------|------|
| | Precision | Recall | Acc. | Precision | Recall | Acc. | Precision | Recall | Acc. |
| **Statistics-based Methods** | | | | | | | | | |
| StyloAI | 21.68 | 22.82 | 23.20 | 37.33 | 37.24 | 41.21 | 75.23 | 72.55 | 77.90 |
| Ullah et al. | 23.83 | 25.29 | 26.01 | 41.05 | 41.04 | 46.41 | 75.91 | 73.83 | 78.59 |
| Binoculars | 21.13 | 22.67 | 24.05 | 40.48 | 39.14 | 43.72 | 72.07 | 74.17 | 77.89 |
| **Model-based Methods** | | | | | | | | | |
| BERT-base | 32.97 | 32.71 | 33.76 | 44.38 | 42.41 | 50.47 | 85.91 | 88.29 | 88.24 |
| ALBERT-base | 32.26 | 31.74 | 31.65 | 41.98 | 40.61 | 44.34 | 83.59 | 79.63 | 84.58 |
| RoBERTa-base | 35.20 | 31.78 | 32.21 | 38.22 | 40.20 | 47.64 | 87.85 | 88.88 | 89.52 |
| Sniffer | 30.63 | 30.77 | 30.04 | 44.39 | 42.88 | 45.65 | 82.55 | 81.30 | 84.45 |
| POGER | 31.28 | 30.16 | 31.88 | 45.41 | 44.68 | 47.16 | 79.17 | 82.03 | 84.89 |
| LLM-Idiosyncrasies | 32.91 | 33.16 | 32.31 | 49.81 | 45.71 | 49.96 | 88.09 | **90.15** | 89.07 |
| **PLAD Framework with Different Classifiers (ours)** | | | | | | | | | |
| PLAD *w/ GBC* | 30.61 | 31.27 | 31.79 | 49.41 | 48.00 | 52.83 | 86.16 | 85.32 | 87.63 |
| PLAD *w/ XGBoost* | 32.11 | 33.51 | 34.04 | **50.73** | **48.77** | **53.30** | 86.45 | 85.31 | 87.79 |
| PLAD *w/ CatBoost* | **35.87** | **36.45** | **36.94** | 50.06 | 47.34 | 51.89 | **88.70** | 87.28 | **89.62** |

**Results.** Table 2 presents the evaluation results across three detection tasks. It can be seen that classifiers with our PLAD framework outperform existing methods on most metrics. For the most challenging *model identification task*, PLAD with CatBoost achieves the best overall performance. This represents a notable improvement over existing approaches. The consistent performance across all metrics suggests that the psycholinguistic features effectively capture distinctive patterns among different LLMs. For the *provider identification task*, PLAD with XGBoost demonstrates the strongest performance. While LLM-Idiosyncrasies shows competitive precision, our approach maintains more balanced performance across all metrics. For the *AIGT classification task*, the results show a more competitive landscape. LLM-Idiosyncrasies achieves the highest recall, while our PLAD framework with CatBoost attains the best precision and accuracy. RoBERTa-base also demonstrates strong performance. The overall performance trend across tasks reflects the challenge of capturing subtle stylistic differences between similar LLMs, highlighting the value of our approach in providing interpretable insights.

## 4.2 ZERO-SHOT EXPERIMENT

Given the observation that model providers often share similar training methodologies, data sources, and architectures across different versions. Therefore, a model family inherits a specific style imprint (Spiliopoulou et al., 2025). In this experiment, we evaluate the generalizability of detection methods on unseen AI models. We design a zero-shot experiment by excluding the latest GPT-o3 and Gemini-2.5 from the training data while retaining other models from their respective providers (OpenAI and Google). This setup mimics the scenario that detection systems encounter newly released models that are not available during training. We compare our PLAD frame-

Table 3: Zero-shot performance comparison on the Provider Identification (6-way) task. Acc. denotes accuracy on the testing set of the seen model.

| Method | Unseen | Acc. | 0-shot Recall |
|--------|--------|------|---------------|
| PLAD | GPT-o3 | 46.46 | **56.34** |
| BERT-base | GPT-o3 | 42.42 | 25.35 |
| PLAD | Gemini-2.5 | 43.72 | **58.46** |
| BERT-base | Gemini-2.5 | 45.23 | 52.31 |

work and fine-tuning BERT-base on the reduced dataset, then evaluate their accuracy on seen models and recall on unseen target models. As results are shown in Table 3, our PLAD framework significantly outperforms BERT in identifying unseen models, demonstrating that the psycholinguistic features extracted by PLAD can capture more robust and generalizable traces of model families. In the context of rapid iteration of LLMs, this detection capability of new models makes our framework more practical than model-based methods.

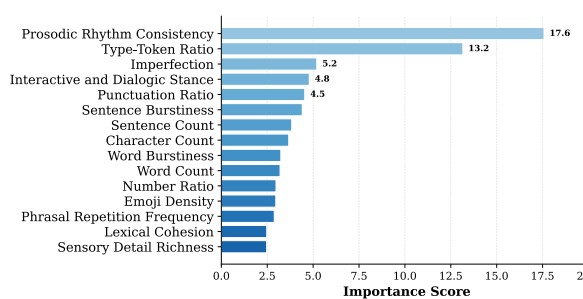

Figure 3: The top-15 important feature scores.

Table 4: Ablation experiment result of each feature dimension. We report the macro-average F1-score (%) on three classification tasks.

| Configuration | 17-way | 6-way | binary |
|---|---|---|---|
| CatBoost (Full) | **36.16** | **48.66** | **87.98** |
| w/o Emotional Dim. | 27.68 | 43.22 | 86.32 |
| w/o Cognitive Dim. | 31.40 | 43.56 | 86.44 |
| w/o Lexical Dim. | 26.81 | 38.66 | 85.00 |
| w/o Cohesion Dim. | 32.29 | 44.76 | 86.57 |

### 4.3 ABLATION AND FEATURE IMPORTANCE STUDY

To understand how PLAD achieves its detection performance, we conduct an analysis combining a dimension-level ablation study and evaluation of feature importance using CatBoost classifier. Figure 3 and Table 4 present the feature importance and ablation study result. It reveals that the largest performance drop occurred when the Lexical Identity and Stylistic Signature features are removed, leading to a substantial decrease across all three tasks. This finding identifies the two most important features, *Prosodic Rhythm Consistency* (17.6) and *Type-Token Ratio* (13.2), suggest that AI-generated text tends to exhibit smoother and more uniform rhythmic patterns, whereas human writing often contains irregularities caused by cognitive processing. The third most important feature, *Imperfection* (5.2), also from this dimension, which detects the absence of human-like hesitations, self-corrections, and other disfluencies that LLMs are optimized to avoid.

In addition, the analysis highlights the significance of Emotional and Social Grounding. Features from this category, such as *Interactive and Dialogic Stance* (4.8) and *Emoji Density* (3.0), rank highly in importance. This demonstrates PLAD's ability to distinguish the subtle markers of human social awareness and interaction from the engagement patterns produced by AI. In summary, PLAD identifies AI content by recognizing its too perfect, too uniform, and lack of social characteristics. We also analyze the differences between human and AI text in Appendix.

## 5 ANALYSIS

In this section, we use PLAD framework to analyze the exploration set in the RedNote Engagement Dataset, uncovering the impact of AI-generated content on user behavior and interaction trends in real-world data with temporal dynamics.

### 5.1 TEMPORAL DYNAMICS OF AI CONTENT

Figure 4 presents the temporal track of AI-generated content proportion over the past 600 days. Despite considerable short-term noise, the overall linear trend ($slope = 0.012$ per day, $0.355$ per month) clearly demonstrates a steady rise in the adoption of AI-authored posts.

In the initial phase, the proportion of AI content remained relatively modest and exhibited substantial volatility. However, from mid-2024 onward, the trend reveals

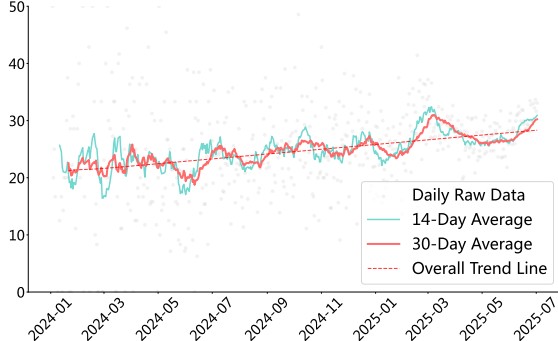

Figure 4: Temporal evolution of AI content proportion in exploration set, where visualizations are smoothed using 14/30-day rolling averages.

a more persistent upward shift, indicating that **AI tools are becoming a significant part of social media activity**. It is also notable that the track does not follow a simple monotonic increase. Several plateaus and temporary declines can be observed (e.g., mid-2024 and mid-2025). These inflection points coincide with platform-level governance measures on AI-generated content, as recorded in the news during the same period.

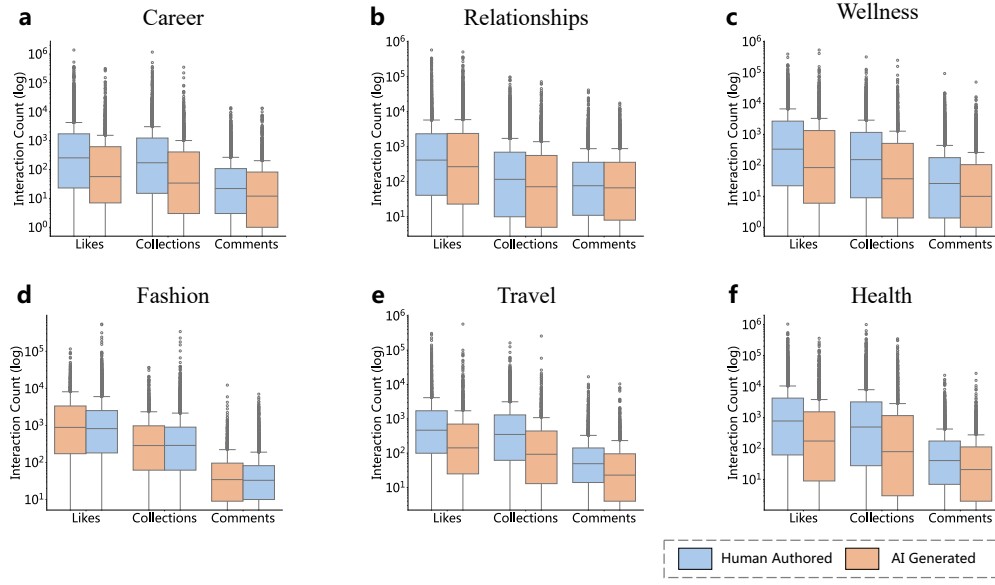

Figure 5: Engagement metrics comparison between Human-authored and AI-generated posts across 6 major topical domains. Human content achieves higher metrics than AI content in most domains.

Overall, the findings indicate that while AI adoption in social media continues to grow steadily in the long term, its short-term dynamics remain subject to both platform governance and stochastic factors.

## 5.2 ENGAGEMENT METRICS OF AI VS. HUMAN POSTS

To investigate engagement disparities, we compare metrics for human-authored and AI-generated posts across the 6 most data-rich domains. Given the heavy-tailed nature of interaction data, we apply a base-10 logarithmic transformation to normalize the distributions. Figure 5 presents these comparisons using boxplots, with blue indicating human content and orange indicating AI content.

The analysis reveals that human-authored posts consistently outperform their AI counterparts, achieving higher median interaction counts and a greater propensity for high engagement (i.e., longer upper whiskers). This gap is particularly visible in domains requiring nuanced personal experience and emotional resonance, such as *Travel*, *Career*, and *Relationships*. This suggests that **current AI content is significantly weaker than human content at establishing emotional resonance**.

Furthermore, compared to human content, the upper whiskers of AI content are shorter. This indicates that the interaction for AI content is more concentrated with less variance. In other words, **AI content tends to be homogeneous and is less likely to produce the exceptionally high-engagement posts**. Conversely, human content's longer upper whisker demonstrates its capacity to produce breakout posts that capture the collective imagination. This virality is often driven by novelty, raw emotional expression, or a unique personal story that breaks a predictable pattern.

Looking into the variations between interaction types, which reveals the cognitive investment from users. For instance, a like is a passive and lightweight social signal, whereas a comment requires active and conscious engagement. Human content shows a steep decline from likes to comments across most domains, while AI content maintains a relatively flatter curve. The Fashion domain presents an exception, showing comparable engagement patterns. These suggest that **AI-generated content, while receiving lower absolute engagement, tends to provoke more discussion relatively**.

## 5.3 AUTHOR-LEVEL AI USAGE AND ENGAGEMENT

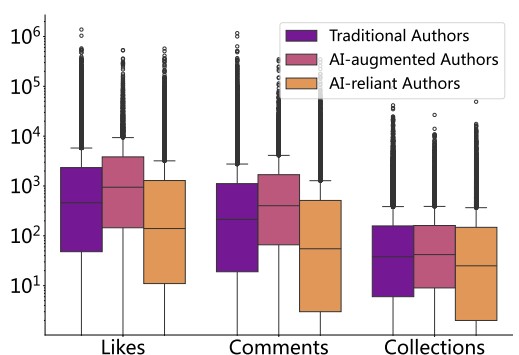

Figure 6: Author-level analysis of AI usage patterns and engagement outcomes.

In addition to content-level analysis, we conduct an author-centric study to understand how different AI usage behaviors correlate with engagement. We select authors with a minimum of four posts to ensure sufficient data for usage calculation, resulting in a total of 829 authors. Based on the proportion of AI-generated content they produced, we categorize these authors into three tiers, which consist of 68.7% Traditional Authors, who compose manually without AI; 27.7% AI-reliant Authors, who fully rely on AI for posting; and a small but distinct group of 3.5% of AI-augmented Authors, who combine human- and AI-generated content to create posts.

As the results illustrated in Figure 6, the Traditional Author and AI-reliant Author groups have similar trends to the analysis in the previous section. However, the AI-augmented Authors group consistently achieves higher engagement across all three metrics. This finding suggests that a balanced, strategic integration of AI tools outperforms approaches that rely exclusively on either human or AI-driven creation. **The most successful authors are leveraging AI while applying human creativity to create more engaging and popular content.**

## 6 RELATED WORK

Social media datasets have been developed for AIGC detection, including TweepFake (Fagni et al., 2021) for early Twitter content, SAID (Cui et al., 2023) for modern LLM detection, and ElectionRumors2022 (Schafer et al., 2024) for election-related content analysis. Chinese social media datasets focus primarily on Sina Weibo for tasks like NER (Peng & Dredze, 2015) and fake news detection (Yang et al., 2021), but lack coverage of RedNote platform and temporal dynamics.

AIGC detection methods span three categories: watermarking techniques that embed imperceptible marks during generation (Kirchenbauer et al., 2023; Liu et al., 2024), classifier-based approaches using fine-tuned transformers like BERT and RoBERTa (Hu et al., 2023; Huang et al., 2024), and statistical methods that establish discrimination thresholds (Mitchell et al., 2023; Zhang et al., 2024). Multi-class detection frameworks like Sniffer (Li et al., 2023) and LLM-Idiosyncrasies (Sun et al., 2025) leverage LLM embeddings for classification. However, existing statistical methods struggle with real-world accuracy (Qiu et al., 2024; Sadasivan et al., 2023), highlighting the need for more robust and interpretable approaches.

## 7 CONCLUSION

In this work, we introduce the RedNote-Vibe, the first comprehensive social media AIGC detection dataset that captures temporal dynamics and engagement patterns. To address the limitations of existing AIGC detection methods in social media contexts, we propose the PsychoLinguistic AIGT Detection Framework (PLAD), which adopts psychological and linguistic theories to achieve both high accuracy and interpretability. Leveraging our dataset and detection method, our analysis reveals several important insights: (1) AI content adoption on social media platforms shows steady growth but is sensitive to platform governance policies; (2) Human-generated content consistently achieves higher engagement, particularly in domains requiring emotional resonance; (3) Authors who combine human creativity with AI assistance achieve the highest engagement levels, suggesting potential benefits of human-AI collaboration. The code and datasets will be publicly available, which provides a new resource for future social media research.

ETHICS STATEMENT

Our dataset is constructed from publicly available content on social media platform, and we have masked all personally identifiable information, including author names, user locations, etc. The AI-generated parallel data is created using paid commercial APIs. This research does not involve any ethical or copyright issues.

REPRODUCIBILITY STATEMENT

To ensure full reproducibility, we will publicly release our dataset, feature extraction and evaluation methodologies, including code implementations and hyperparameter settings after publication. We have uploaded dataset examples in the supplementary material.

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

# APPENDIX

In the Appendix, we provide further analysis, including the differences between human-written text and AI-generated text from the perspective of the PLAD framework (Section A.1) and how these features affect the engagement metrics of posts (Section A.2). In addition, we analyze the correlation between features, reflecting the orthogonality of our framework (Section A.3). Finally, we introduce details of our AI samples generation (Section B.1) and feature extraction method (Section B.2). We also provide a declaration of LLM usage in Section C.

# A    FURTHER ANALYSIS

## A.1    FEATURE STATISTICS FOR HUMAN AND AI POST

To understand the differences in text style between human and AI, we conduct a comparative analysis of linguistic features extracted from human- and AI-authored texts. We apply statistical tests across the dataset, and we select six representative features for illustration, covering the four dimensions of our framework. As shown in Figure 7, these features highlight systematic divergences between human and AI writing styles.

The results indicate that AI texts consistently achieve higher values in imperfection, reflecting stable fluency and a lack of surface-level flaws. By contrast, human writing displays a much broader distribution. A similar contrast appears in prosodic rhythm consistency and sentence burstiness. AI-generated texts demonstrate regularity and uniformity in rhythm and fluctuations in sentence patterns, whereas human writing is more dynamic and irregular, often breaking rhythmic patterns. In the lexical level, the type-token ratio results show that AI text tends to maintain higher lexical diversity, reflecting stochastic generation processes that avoid

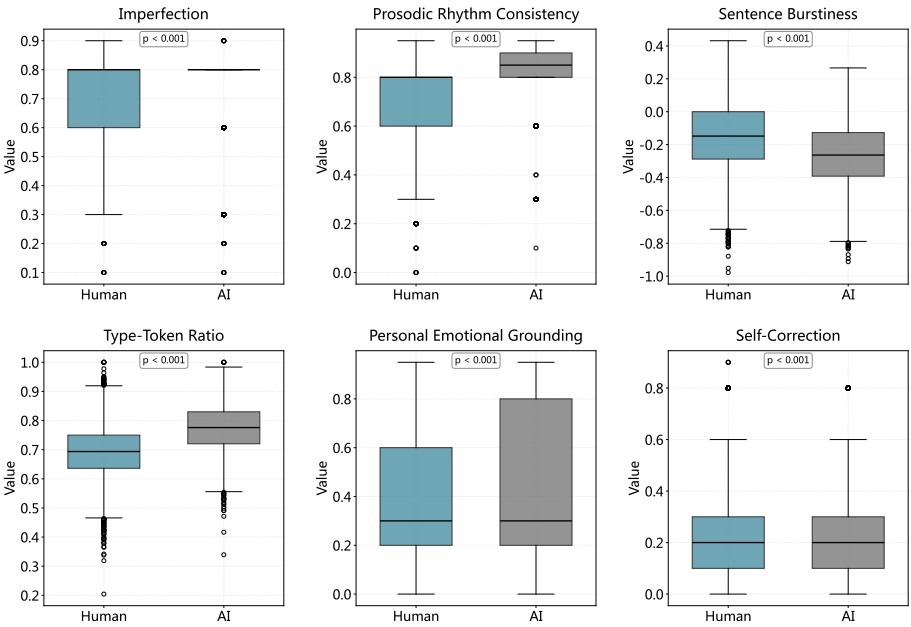

Figure 7: Comparison of feature statistics between human and AI-generated texts.

frequent repetition. In contrast, human writers employ more shallow lexical distributions, which is shaped by personal stylistic habits.

In terms of emotion and rhythm, AI can achieve higher scores than human writing. This suggests that AI can mimic or even outperform humans in many human-like writing characteristics, while real human writing is more restrained.

## A.2 FEATURE ANALYSIS FOR ENGAGEMENT

To discover the relationship between features and engagement metrics (i.e. `likes`, `collections`, and `comments`), we conducted an analysis using SHAP value. For each metric, we train a `CatBoost` regressor model, denoted by $f(x)$, using its default hyperparameters. Subsequently, we employ SHapley Additive exPlanations (SHAP) to interpret the model's predictions (Lundberg & Lee, 2017). SHAP attributes an importance value to each feature based on principles from cooperative game theory, ensuring properties like local accuracy and consistency.

As shown in Figure 8, we observe distinct patterns of feature influence across different engagement types. For "likes", the most salient predictors are *Punctuation Ratio* and *Word Frequency Entropy*. This indicates that lightweight forms of engagement are primarily driven by surface-level features such as punctuation density and lexical diversity. In contrast, higher-order psycholinguistic features (e.g., *Perspectival Complexity*, *Axiological Coherence*) play only a secondary role. This suggests that likes are largely sensitive to readability and rhythm rather than deeper cognitive or semantic structures.

In the case of "collections", *Word Frequency Entropy* emerges as the dominant feature, followed by *Phrasal Repetition Frequency* and *Axiological Coherence*. Compared with likes, collections are more strongly associated with content richness and value consistency. *Axiological Coherence* further suggests that users are more inclined to preserve texts that demonstrate coherent values and internal logical alignment. Thus, collections appear to reflect more deliberate and evaluative forms of engagement.

For "comments", in addition to *Punctuation Ratio*, socially oriented features such as *Lexical-Stylistic Personalization*, *Empathetic Engagement*, and *Interactive and Dialogic Stance* exhibit the strongest influence. Unlike likes or collections, commenting behavior is primarily shaped by interpersonal dynamics, empathy, and argumentative stance. This highlights the role of dialogic and relational features in fostering deeper interactions.

The details of calculating SHAP value are shown as follow:

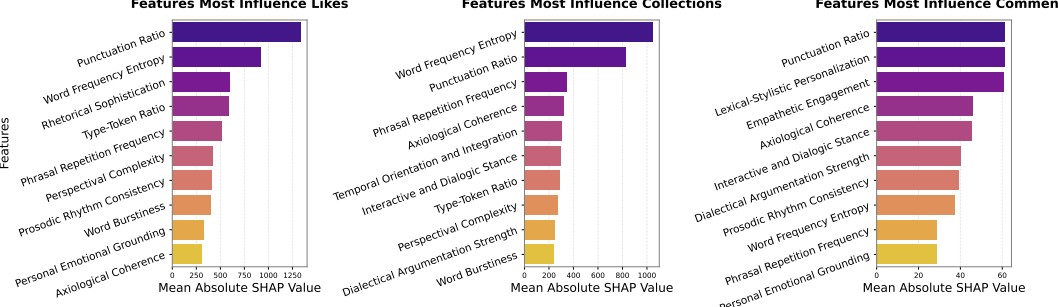

Figure 8: Top-10 most influential features of likes, collections and comments metrics.

For a single prediction $f(x)$, SHAP explains it using an additive feature attribution model, $g(x')$:

$$g(x') = \phi_0 + \sum_{i=1}^{M} \phi_i x'_i$$

where $g(x') \approx f(x)$, $x'$ is a simplified binary input representing the presence ($x'_i = 1$) or absence ($x'_i = 0$) of a feature, $M$ is the number of features, and $\phi_i \in \mathbb{R}$ is the SHAP value for feature $i$. The term $\phi_0 = E[f(x)]$ represents the base value, which is the mean prediction over the exploring set.

The SHAP value $\phi_i$ for each feature is calculated as its marginal contribution to the prediction, averaged across all possible feature orderings (coalitions), and is formally defined as:

$$\phi_i = \sum_{S \subseteq F \setminus \{i\}} \frac{|S|!(|F| - |S| - 1)!}{|F|!} [f_x(S \cup \{i\}) - f_x(S)]$$

where $F$ is the full set of features, $S$ is a subset of features not including $i$, and $f_x(S)$ is the model's expected output conditioned on the feature values in $S$. For our analysis, we used the mean absolute SHAP value, $\frac{1}{N} \sum_{j=1}^{N} |\phi_i^{(j)}|$, as the metric for global feature importance.

### A.3 FEATURE CORRELATION ANALYSIS

To ensure that the proposed features capture complementary aspects of text, we conducted a pairwise correlation analysis. Figure 9 illustrates the distribution of Pearson correlation coefficients across all feature pairs ($N = 465$).

Overall, the results confirm that the features are weakly correlated. The average absolute correlation is $0.1884$ with a standard deviation of $0.2290$, suggesting that the majority of features contribute orthogonal information.

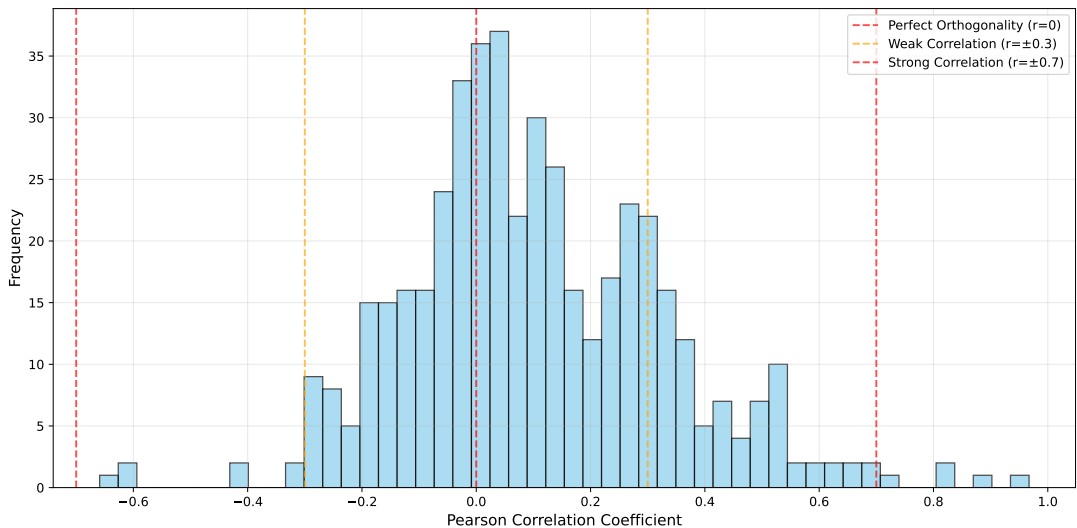

Figure 9: Distribution of pairwise feature correlations. Dashed lines indicate thresholds for weak ($|r| = 0.3$) and strong ($|r| = 0.7$) correlation. Most feature pairs are weakly correlated, demonstrating orthogonality of the feature set.

Specifically, only 6 feature pairs exhibit strong correlations ($|r| > 0.7$), while 89 pairs fall into the medium range ($0.3 < |r| < 0.7$). The vast majority, 370 feature pairs, remain weakly correlated ($|r| < 0.3$).

This distribution indicates that the feature set avoids redundancy and is well-suited for capturing distinct dimensions of linguistic behavior. Therefore, the framework benefits from a diversified set of signals spanning emotional grounding, cognitive architecture, stylistic identity, and textual cohesion. It provides a robust foundation for interpretable AIGT detection.

# B    IMPLEMENTATION DETAILS

## B.1    AI SAMPLE GENERATION DETAILS

To ensure reproducibility and transparency, we provide implementation details of our AI sample generation pipeline. The generation process is controlled by a seed-based prompting strategy, where each human-authored note serves as the reference input. Given a seed note and its corresponding domain label, we construct a structured prompt (see pseudocode below) that instructs the LLM to produce a new RedNote-style post. The prompt enforces the following constraints: (1) emulate the stylistic and colloquial properties of the seed (including informal punctuation and minor grammatical imperfections common in social media), (2) maintain thematic and length consistency with the seed, while ensuring semantic novelty, and (3) output results strictly in a predefined JSON schema containing the `title` and `content` fields.

```
Prompt(seed_note, domain):
  "Reference snippet: {seed_note}

  Please create a new RedNote-style post based on the
  reference above. Requirements:
  1. The new post should have a similar theme and topic
     domain ({domain}) but must not be identical to the
     reference.
  2. Mimic the writing style of the reference, including
     colloquial tone, informal punctuation, and possible
     minor errors typical of social media.
  3. Keep the length roughly consistent with the seed note.
  4. Output strictly in JSON format as follows:

  {
    "title": "Post title",
    "content": "Post content"
  }

  Output only the JSON object, without any extra text."
```

Figure 10: Pseudocode of the prompt used for AI sample generation.

We employ commercial paid APIs from multiple providers to generate the AI samples. The generation script incorporates error handling, including automated detection of extraneous markdown wrappers (e.g., "```json) and recovery via JSON string extraction. Invalid or unparsable generations are discarded to ensure dataset integrity.

Each LLM in our pool receives identical prompts and seed distributions, guaranteeing fairness across providers. Importantly, we only select seed notes published prior to November 2022 to exclude potential AI-generated contamination.

## B.2 FEATURE EXTRACTION

We develop an automated pipeline to extract features from each text entry using a standardized LLM-based scoring system. Each feature is defined through structured JSON criteria that specify evaluation dimensions, scoring rubrics, and key indicators. The features are listed in Table 5. We use the latest qwen-turbo (2025-07-15) as the proxy model.

### B.2.1 SCORING FRAMEWORK

Our approach employs a two-stage process: (1) text preprocessing involving removal of extraneous characters and normalization, and (2) LLM-based evaluation using dynamically assembled prompts. Each of the 31 features is defined by a JSON schema containing:

- **Dimension description**: Definition of the psychological construct.
- **Scoring criteria**: Anchored 0-1 scale with explicit behavioral markers.
- **Key indicators**: Textual evidence to focus evaluation.
- **Few-shot Examples**: A set of text samples paired with their expert-assigned scores. These examples guide the model's in-context learning, calibrating its judgment to align with human evaluation standards.

For each text sample, we dynamically construct evaluation prompts by embedding the target text and relevant feature criteria into a standardized template that instructs the LLM to follow a Chain-of-Thought reasoning process.

### B.2.2 PROMPT TEMPLATE STRUCTURE

The evaluation prompt is dynamically constructed by assembling five core components into a standardized template. It begins by establishing the Task Context to define the psycholinguistic analysis objective, which identifies the evaluation task. The template then assigns a Role Definition, positioning the LLM as an expert evaluator. Subsequently, the specific Dimension Specification is injected from the JSON file, followed by the preprocessed Target Text for evaluation. Finally, the prompt provides detailed CoT Instruction, guiding the LLM through reasoning steps with metacognitive checks to ensure a rigorous scoring process.

The LLM is instructed to output only a numerical score between 0.0 and 1.0, ensuring standardized quantitative assessment across all features.

### B.2.3 EXAMPLE: EMOTIONAL INTENSITY

To illustrate our approach, we present the JSON example for emotional intensity evaluation:

```
{
  "dimension_id": "emotional_intensity",
  "description": "Evaluates the depth, regulation, and contextual
      appropriateness of emotional expression. This dimension assesses the
      presence of emotion, and its nuance, variability, and alignment with the
      narrative events. High scores reflect a rich, well-regulated, and
      contextually congruent emotional landscape, while low scores indicate
      expressions that are flat, extreme, or mismatched with the situation.",
```

```
  "scoring_criteria": {
    "0_score": "Emotionally flat, suppressed, or chaotically unregulated. The
        expression is either monotonous (e.g., alexithymic, detached) or
        extreme and overwhelming (e.g., hysterical, disproportionate rage).
        There is a significant incongruence between the emotion described and
        the context.",
    "1_score": "Rich, nuanced, and contextually appropriate emotional
        expression. The author conveys a spectrum of feelings using diverse
        vocabulary. Emotional intensity is well-regulated, rising and falling
        in a way that is congruent with the narrative. Acknowledges complex or
        mixed emotions."
  },
  "key_indicators": [
    "Analyze the ratio and distribution of positive (e.g., 'joy', 'relief') vs.
        negative (e.g., 'grief', 'fear') emotion words. Assess the mix of high
        -arousal (e.g., 'ecstatic', 'furious') vs. low-arousal (e.g., 'serene',
        'content') terms.",
    "Evaluate the richness of the emotional lexicon. Does the author use a
        variety of synonyms and descriptors for feelings, or repeatedly use the
        same basic emotion words?",
    ...
  ],
  "few-shot examples": [
    {
      "text": "I can't believe she left. I'M SO ANGRY! EVERYTHING IS AWFUL! I
          will NEVER be happy again, this is the worst thing that could ever
          happen to anyone! I hate everything and everyone!",
      "score": 0.3,
      "rationale": "While strong emotion is present, it is extreme, one-
          dimensional, and unregulated. The use of absolutes ('NEVER', '
          EVERYTHING') and disproportionate intensity without nuance suggests a
           lack of emotional modulation, mapping to the lower end of the scale
          ."
    },
    ...
  ]
}
```

This structured approach ensures consistent, objective evaluation across all 31 psycholinguistic dimensions while maintaining the flexibility to adapt criteria for different psychological constructs.

Table 5: Detailed feature list of the PLAD framework.

| Dimension | Features | References |
|---|---|---|
| Emotional and Social Grounding | Emotional Intensity | Gross 1998 |
| | Personal Emotional Grounding | Conway & Pleydell-Pearce 2000 |
| | Sensory Detail Richness | Levine et al. 2002 |
| | Social Connectedness | Giles & Powesland 1975 |
| | Empathetic Engagement | Baron-Cohen 1997 |
| | Interactive and Dialogic Stance | Bakhtin 2010 |
| | Unique Emoji Ratio | Felbo et al. 2017 |

*(Continued on next page)*

Table 5: (continued)

| Dimension | Features | References |
|-----------|----------|------------|
| | Emoji Density | Felbo et al. 2017 |
| Cognitive Complexity and Worldview Integration | Perspectival Complexity | Baker-Brown et al. 1992 |
| | Narrative Structure Flexibility | Labov & Waletzky 1997 |
| | Dialectical Argumentation Strength | Toulmin 2003 |
| | Self-Correction | Hyland 2005 |
| | Axiological Coherence | Graham et al. 2009 |
| | Temporal Orientation and Integration | Reichenbach 1947 |
| | Sentence Count | Manning & Schutze 1999 |
| | Word Count | Manning & Schutze 1999 |
| | Character Count | Manning & Schutze 1999 |
| Lexical Identity and Stylistic Signature | Lexical-Stylistic Personalization | Argamon et al. 2003 |
| | Prosodic Rhythm Consistency | Halliday & Matthiessen 2013 |
| | Imperfection | Clark & Tree 2002 |
| | Rhetorical Sophistication | Grice 1975 |
| | Punctuation Ratio | Manning & Schutze 1999 |
| | Number Ratio | Manning & Schutze 1999 |
| | Type-Token Ratio | Templin 1957 |
| | Word Frequency Entropy | Shannon 1948 |
| | Word Burstiness | Gries 2008 |
| Cohesion and Repetition | Lexical Cohesion | Halliday & Hasan 2014 |
| | Inter-Sentential Sentence Similarity | Foltz et al. 1998 |
| | Immediate Repetition Density | Stamatatos 2009 |
| | Phrasal Repetition Frequency | Stamatatos 2009 |
| | Sentence Burstiness | Gries 2008 |

## C LLM USEAGE

In this work, LLMs are used in two scenarios. Firstly, we use LLMs as an auxiliary tool for grammatical checking and language polishing to improve the clarity and readability of the manuscript. The core contributions, including the research ideas, experimental design, and data analysis, are conducted **without** the involvement of LLMs.

Secondly, LLMs are used for data generation and evaluation. All specific LLM usages are explicitly detailed in the main sections and the appendix of this paper.

