# OpenReview forum: "RedNote-Vibe: A Dataset for Capturing Temporal Dynamics of AI-Generated Text in Social Media"
_ICLR.cc/2026/Conference — ICLR 2026 Conference Withdrawn Submission_

### Official Review · Reviewer_t6i2 · 2025-10-31

**Soundness:** 2
**Presentation:** 3
**Contribution:** 2
**Rating:** 2
**Confidence:** 5

**Summary:**

This paper looks at how to spot AI-generated text in dynamic social media environments. It introduces the PsychoLinguistic AIGT Detection Framework (PLAD) — an interpretable method that uses psycholinguistic features. Not only does it deliver strong performance using model-based techniques, but it also gives clear insights into the style fingerprints of different LLMs and shows how AI-generated text connects to user engagement. By digging deep into the dataset, the study uncovers trends in AI adoption over time and highlights differences in engagement patterns between human-written and AI-generated content.

**Strengths:**

1. This work focuses on the social media domain, capturing a large-scale real-world dataset from RedNote and constructing a parallel version with AI-generated content. The dataset is intentionally designed to support research on AI-generated discourse (AIGD).

2. From an interpretable perspective, the study develops a framework based on psycholinguistic features, which is used for training and predicting with AI content detectors.

**Weaknesses:**

1. The paper lacks sufficient details about the dataset construction process.


   - Line 078: Please clarify the source and location of RedNote’s official user behavior report.

   - Line 084: What are the filtering criteria? Is the metadata from raw crawled data, or is it already processed after filtering?

   - Please provide a clear description of the language distribution of the dataset.

   - Regarding the parallel AI-generated posts: they appear to be rewrites rather than entirely original creations, as the original content is supplied and then rewritten based on prompts. If so, please explain the rationale and methodology behind this process.

   - Line 117: What are the “6 providers”? Are they specific companies? Please elaborate.


2. As the core contribution of the paper, the description of the psycholinguistic features is overly superficial.

   - In Section 3, the “four dimensions of human language expression” are only defined in terms of what aspect they measure, without listing the 31 linguistic features in detail or explaining how each feature is extracted.

   - The appendix only shows six representative features for illustration, with comparative results, making it hard to understand the implementation and reproduction process.

   - Please describe the feature extraction methodology clearly, especially for different languages — are the extraction methods identical across languages? The paper does not address this point.


3. Concerns About Experimental Reliability

   - Some average sequence lengths exceed 512 tokens, yet several PLM-based models used in the paper have a maximum input length of 512. Was the input simply truncated? If so, please clarify. Considering alternatives like Longformer could be beneficial [1][2].

   - Previous works [1] found that using a small set of features as model input did not surpass PLM-based methods in accuracy. This paper uses 31 features, which sounds promising, but the importance and justification of this choice are not clearly discussed.

   - The paper does not provide enough details on dataset splits (training/validation/test).

   - Prior studies have reported PLM-based methods reaching >98% accuracy [1][2]; however, the paper’s reported accuracy is below 90%, raising questions about the credibility of the results.

   - For Subsection 4.2 regarding zero-shot experiments, prior works [1][2][3] suggest that PLM-based models are not as weak in cross-source detection as the reported results here. This discrepancy needs explanation.


4. Although the **Ethics Statement** briefly mentions masking personally identifiable information, the dataset construction section does not explain how privacy protection was implemented. A more comprehensive description is necessary.

5. While the paper claims to focus on AI detection in dynamic data, the dataset itself is static. The time-related analysis is only performed through the “Exploration Set.” This is similar in spirit to [3], except the domain of data differs.

6. Some models used (e.g., Claude, Gemini) are not commonly used in the Chinese internet context. This could introduce bias in the training data. Using such models to train and then evaluate predictions on real-world data may weaken the validity of conclusions.

[1] Beyond Binary: Towards Fine-Grained LLM-Generated Text Detection via Role Recognition and Involvement Measurement

[2] MAGE: Machine-generated Text Detection in the Wild

[3] Large language models penetration in scholarly writing and peer review

**Questions:**

See weakness.

015：Xiaohongshu? maybe rednote?

**Details Of Ethics Concerns:**

Although the **Ethics Statement** briefly mentions masking personally identifiable information, the dataset construction section does not explain how privacy protection was implemented. A more comprehensive description is necessary.

---

> ### Author Response · Authors · 2025-11-17
>
> To Reviewer t6i2,
>
> We thank you for your detailed review. While we appreciate the time you took, we believe the low rating stems from several misunderstandings of our paper's methodology and contribution. We will address each of your concerns directly.
>
>
> **W1: Details on dataset**
>
> *Q1.1: Report Source*
>
> [1] http://xhslink.com/o/3ZtMUSbgoCl
>
> This report is an official post from the platform's official account, identifying the 10 dominant content categories.
>
> *Q1.2: Filtering Criteria*
>
> We collect 120k notes via tag search, but some posts are ads/non-text or cross-domain. We filter out notes that are not strongly align with target categories.
>
> *Q1.3: Language*
>
> RedNote is a Chinese social media platform. As stated in the paper (L70, L402), all content in our dataset is in Chinese.
>
> *Q1.4: AIGT methodology*
>
> We would like to clarify our AI sample generation process. We intentionally use a seed-based approach (L112, Appendix B.1) where the human post serves as a reference for style, tone, and topic. In our preliminary experiments, we tried different methods and found that this approach can maximize the challenge of the detection and can generate posts that most closely resembled human creation.
>
> *Q1.5:What are the “6 providers”?*
>
> The 6 providers are listed in our data construction pipeline in Figure 2.
>
> **W2: Details on features**
>
> We are concerned by the claim that our features are "overly superficial" due to featres are "not listed" and "only shows six features for illustration."
>
> *Q2.1-2: Missing the 31 features;*
>
> Due to the page limit, we explicitly state in the main text (L216) that the full list is in the appendix. Appendix Table 5 provides the complete list of all 31 features, their corresponding dimension, and their theoretical grounding with academic references.
>
> *Q2.3: feature extraction*
>
> The feature extraction method is presented in Appendix B.2. As all data is Chinese, our feature extraction is designed for Chinese. The appendix provides an English prompt example for reviewer clarity, but as we state, a cross-lingual LLM is used for extraction, making the prompt's language irrelevant to its performance on the Chinese text.
>
> **W3: Concerns About Experimental Reliability**
>
> *Q3.1: Was the input truncated?*
>
> Yes, inputs longer than 512 tokens were truncated. This is a standard and necessary practice for BERT-based classification models and follows common setup. We will add an explicit sentence stating this in the revision.
>
> *Q3.2: 31 Features vs. PLM*
>
> The core novelty of PLAD is not the quantity of features, but their psycholinguistic foundation. These features are designed from established psychological theories to capture why AI text "feels" different. Our ablation study (Fig. 3, Table 4) and feature analysis (Appendix A.1-3) demonstrate their importance and orthogonality. The experimental results, which show PLAD outperforming PLM-based methods, validate this new approach.
>
> *Q3.3: Data Splits*
>
> We apologize for this omission. We used a standard 8:1:1 split for training, validation, and testing. We will add this to the revision.
>
> *Q3.4: Low Accuracy of PLMs < 90%*
>
> We must argue that this comparison highlights a key challenge. Model performance is highly dataset-dependent. The >98% accuracy cited in other work is on different, and might be less challenging datasets. As explained in W1 (Q1.4), we designed our dataset to be difficult, mimicking real-world human-AI stylistic similarity. Our <90% baseline results do not indicate low credibility. Instead, they indicate that **our dataset reflects the true, challenging nature of AIGT detection in the wild.**
>
> *Q3.5: Zero-shot*
>
> The cited works [1][2] compare PLMs to other PLMs on cross-context tasks, not against feature-based methods. On our more challenging dataset, all models show lower generalization, but PLAD's psycholinguistic features capture more robust, generalizable traces than the patterns learned by PLMs. We have expanded our zero-shot comparison, which confirms PLAD’s superior robustness:
>
> | **Method** | **Unseen** | **Acc.** | **0-shot Recall** |
> |---|---|---|---|
> | PLAD | GPT-o3 | 46.46 | **56.34** |
> | ALBERT-base | GPT-o3 | 42.73 | 28.31 |
> | LLM-Idiosyncrasies | GPT-o3 | 44.34 | 22.54 |
> |---|---|---|---|
> | PLAD | Gemini-2.5 | 43.72 | **58.46** |
> | ALBERT-base | Gemini-2.5 | 42.61 | 50.00 |
> | LLM-Idiosyncrasies | Gemini-2.5 | 41.71 | 53.85 |
>
> *W4: Ethics*
>
> Our dataset directly follows standard ethical practices for related social media research.
>
> *W5: Dataset novelty*
>
> We emphasize RedNote-Vibe's core novelty lies in its interaction data metadata. *Interaction count provides the weight of text on its impact.* Existing datasets treat all text with equal weight, but social media data is actually ranked, with higher-traffic posts more likely to be recommended to more people. Our dataset offers the possibility of analyzing how AIGT content can influence humans, rather than simply appearing on social media platforms.

---

### Official Review · Reviewer_kgFj · 2025-10-31

**Soundness:** 2
**Presentation:** 2
**Contribution:** 2
**Rating:** 4
**Confidence:** 3

**Summary:**

This paper presents a real AIGT dataset from social media spanning 5 years and develops an AIGT detector based on psycholinguistic features.

**Strengths:**

1. A 5-year AIGT dataset with a large amount of real data is released, and some analysis of the dataset is provided.
2. PLAD demonstrates superior detection performance.

**Weaknesses:**

1. I believe temporal analysis of the data is a very important contribution. Unfortunately, in Section 5.1, the authors only analyze frequency. Several important analyses are missing: How has AIGT data changed over the 5 years? How has the detection difficulty evolved?
2. The authors lack some basic statistical information about the dataset. For example, what is the yearly frequency distribution of the data?
3. An important issue concerns data bias. The authors only state that “We adopt a web crawler to collect 120,000 notes from January 2020 to July 2025,” but provide no measures to ensure the data is unbiased.
4. The authors construct data through customized LLM simulations. However, a more important issue is how to obtain genuinely published AIGT data from social media.
5. The data construction seems inconsistent with their claimed 5-year span. In line 118, they actually use only data before November 2022 (the pre-LLM period).

**Questions:**

I suggest the authors to include some specific data examples in the paper.

---

> ### Author Response · Authors · 2025-11-17
>
> To Reviewer kgFj,
>
> We sincerely thank you for your time and for raising several important questions.  We would like to address your concerns below, particularly regarding the temporal analysis and dataset construction.
>
>
> **W1, W2, W3: Temporal Analysis & Missing Statistics & Data Bias**
>
> This is a very insightful point, and we appreciate the opportunity to clarify our methodology and the nuances of the dataset.
>
> Our data source was collected at specific time points (i.e., in 2022.11 and 2025.07). As is common with web crawlers, the retrieval process at any given point favors more recent and popular content. This results in our data being denser near the time of data collection and sparser further away. We have updated the timestamp distribution by quarter as follows.
>
> | Season | Samples | % |
> |------|--------|---------|
> | 2020Q4 | 3,274 | 3.19% |
> | 2021Q1 | 6,714 | 6.55% |
> | 2021Q2 | 6,946 | 6.78% |
> | 2021Q3 | 8,301 | 8.10% |
> | 2021Q4 | 7,793 | 7.60% |
> | 2022Q1 | 7,332 | 7.15% |
> | 2022Q2 | 8,092 | 7.89% |
> | 2022Q3 | 10,584 | 10.32% |
> | 2022Q4 | 6,641 | 6.48% |
> | 2023Q1 | 880 | 0.86% |
> | 2023Q2 | 1,173 | 1.14% |
> | 2023Q3 | 1,332 | 1.30% |
> | 2023Q4 | 1,317 | 1.28% |
> | 2024Q1 | 2,683 | 2.62% |
> | 2024Q2 | 2,408 | 2.35% |
> | 2024Q3 | 3,491 | 3.41% |
> | 2024Q4 | 3,157 | 3.08% |
> | 2025Q1 | 6,578 | 6.42% |
> | 2025Q2 | 12,743 | 12.43% |
> | 2025Q3 | 1,075 | 1.05% |
>
> Due to this non-uniform, real-world collection artifact, conducting a statistically day-by-day temporal analysis for 2023 was not feasible. This is precisely why we limited our temporal analysis in Section 5.1 to the last 1.5 years, where the data is sufficiently dense to reveal a clear and reliable trend of AIGT adoption.
>
> For potential data bias issue, as detailed in Section 2.1, our collection was carefully guided to mitigate bias. We follow the latest RedNote platform user behavior report [1], which identifies 10 most dominant content categories on RedNote. This report provided a few example popular tags (search keywords) for each category. For instance, the 'Relationships' category included seed tags like "Couples going Dutch," "How couples divide chores," and "First meeting with a blind date." We then manually curated and expanded this list to approximately 50 representative tags per category to ensure comprehensive coverage and avoid sampling bias from just a few tags. This expanded set of tags are then used as queries to retrieve sample from our data source. We will make this stratified sampling approach clearer in our revision.
>
> **W4, W5: Data construction through LLM simulations; inconsistent with 5-year span.**
>
> We agree that a more important issue is how to obtain genuinely published AIGT data from social media. This is a critical point that highlights the core challenge for the entire AIGT detection field. However, this is an inherently unannotatable task because whether a post was published using AI is unknown to anyone except the original author.
>
> This is why we designed our dataset and divide into three parts:
>
> + Human-Generated Set: 100% confirmed human (2020.01 - 2022.12)
> + AI-Generated Set: 100% confirmed AI (our synthetic variants)
> + Exploration Set: Real-world, post-LLM data with unknown labels (2023.01 - 2025.07)
>
> For data construction span inconsistency, this appears to be a misunderstanding of our dataset's structure, we apologize for this confusion. 5-year span refers to the total collection period for our posts. We intentionally used only human posts from before November 2022 as "seeds" to create our AI-Generated Set.
>
> The full 5-year span of data is indeed presented in the dataset.
>
> **Q1: include some specific data examples in the paper**
>
> We have submitted data examples in Supplementary Material, we will give examples in the revision.
>
> [1] http://xhslink.com/o/3ZtMUSbgoCl

---

### Official Review · Reviewer_MZRp · 2025-10-31

**Soundness:** 4
**Presentation:** 4
**Contribution:** 3
**Rating:** 4
**Confidence:** 4

**Summary:**

The authors introduce RedNote-Vibe, a 5-year social-media dataset from Xiaohongshu (RedNote) with post text plus rich engagement metadata (likes, comments, collections) spanning pre-LLM to July 2025, expressly to study temporal dynamics of AI-generated text (AIGT) in the wild. They also propose PLAD -- a psycholinguistics-based, interpretable detector for AIGT.

**Strengths:**

1. Presentation: It was very easy to read and follow. I appreciate the authors' effort for their clear presentation.
2. Longitudinal, engagement-aware dataset: I like the way the authors combined pre-LLM and post-LLM contents, and framed its usability for AI-content detection and analysis.
3. Interpretable detector (PLAD): The paper proposes a psycholinguistics-based, feature-driven approach aimed at interpretability rather than a purely black-box classifier.

**Weaknesses:**

1. Some subsections need more explanation/details. For example, the 'Data Collection' subsection should be clearer, the 'Feature Extraction and Classification' subsection should add more details, the 'Experiment Setup' subsection should mention how the other baselines were implemented/tweaked for specific identification, etc.

2. For the zero-shot experiment, I would recommend adding more detectors as baselines; only BERT-base is not enough.

3. For the 'Analysis' section, authors should explicitly mention how they are identifying the AI-text and AI-augmented authors. It's not clear from the paper in its current state.

4. There are prior works that covered AI-text in social media, such as [1], Reddit posts in [2], etc. The author needs to clarify how their dataset differs from these existing datasets and why PLAD adds more value to the community.

4. There are some related works that might be worth mentioning, such as [3,4] for AI-augmented users, etc.

References:

[1] Are We in the AI-Generated Text World Already? Quantifying and Monitoring AIGT on Social Media

[2] RAID: A Shared Benchmark for Robust Evaluation of Machine-Generated Text Detectors

[3] Almost AI, Almost Human: The challenge of detecting AI-polished writing

[4] LLM-as-a-coauthor: The challenges of detecting LLM-human mixcase

**Questions:**

1. I did not understand some of the parts in the 'Data Collection' subsection, e.g., *"We first extract the example tags provided in the report for each category, then expand them to approximately 50 representative tags per topic through manual curation."*, etc. Can you elaborate it?
2. How were the statistics-based and model-based methods adopted for the model- and provider- identification task?
3. Is there any specific reason to include only the BERT-base model for zero-shot comparison?
4. For the 'Analysis' section, how are you deciding/filtering the AI-texts?
5. For the subsection 'Author-Level AI Usage and Engagement', how did you identify the AI-augmented authors?

**Details Of Ethics Concerns:**

Need to evaluate if the web-crawled dataset contains any personally identifiable information.

---

> ### Author Response · Authors · 2025-11-17
>
> To Reviewer MZRp,
>
> We sincerely thank you for your detailed feedback and positive assessment of our paper's soundness and presentation. We address each of your points below.
>
> **W1, Q1, Q2: Some subsections need more explanation/details.**
>
> Thank you for pointing out. We have listed some details of data collection and experiment setup below. The details of feature extraction are also provided in Appendix B.2. We will update these sections in the revision.
>
> **Q1: Data Collection details**
>
> One of the features of the RedNote platform is that users can obtain posts based on tag search. We follow the user behavior report 2024 [1], which identifies 10 most dominant content categories on RedNote. This report provided a few example popular tags (search keywords) for each category. For instance, the 'Relationships' category included seed tags like "Couples going Dutch," "How couples divide chores," and "First meeting with a blind date." We then manually curated and expanded this list to approximately 50 representative tags per category to ensure comprehensive coverage and avoid sampling bias from just a few tags. This expanded set of tags is then used as queries to retrieve sample of posts for each category.
>
>
> **Q2: How were the statistics-based and model-based methods adopted for the model- and provider- identification task?**
>
> For the statistics-based methods, we use their extracted feature vectors to train multi-class classifiers with 6 and 17 output classes, respectively. For model-based methods, we replaced their final binary classification layer with a multi-class classification layer (with 6 or 17 output heads). We then fine-tuned the entire model on our training set for the respective multi-class task.
>
> **W2, Q3: Need for more baselines in zero-Shot experiment**
>
> Our initial inclusion of only BERT-base was due to page constraints, as it is a highly representative and widely used baseline. We have update zero-shot experiment results with other strong baselines.
>
> | **Method** | **Unseen** | **Acc.** | **0-shot Recall** |
> |---|---|---|---|
> | PLAD | GPT-o3 | 46.46 | **56.34** |
> | BERT-base | GPT-o3 | 42.42 | 25.35 |
> | ALBERT-base | GPT-o3 | 42.73 | 28.31 |
> | LLM-Idiosyncrasies | GPT-o3 | 44.34 | 22.54 |
> |---|---|---|---|
> | PLAD | Gemini-2.5 | 43.72 | **58.46** |
> | BERT-base | Gemini-2.5 | 45.23 | 52.31 |
> | ALBERT-base | Gemini-2.5 | 42.61 | 50.00 |
> | LLM-Idiosyncrasies | Gemini-2.5 | 41.71 | 53.85 |
>
> **Q4:For the 'Analysis' section, how deciding/filtering the AI-texts?**
>
> We use our proposed PLAD to classify the AI-text in Exploration Set. As reviewer dnne raised concern about circularity of this setting, please allow us to clarify our methodology.
>
> Note that the Exploration Set is inherently unannotable because whether a post was published using AI is unknown to anyone except the original author. Since we cannot obtain ground-truth labels, we apply the best-performing method that we validated in previous section.
>
> Therefore, our analysis in Section 5 is not an evaluation of PLAD. Rather, it is the application of this method to investigate unknown, real-world data and uncover emerging trends. This also provides a baseline and data source for future AI-related social science research.
>
> **W3, Q5: How to identify the AI-relient and AI-augmented authors.**
>
> In the exploration set, we first selected authors who had published multiple blog posts (>4). Then, we categorized these authors based on the classification results of their blog posts:
>
> + Traditional Authors: 0% of their posts were classified as AI.
>
> + AI-reliant Authors: 100% of their posts were classified as AI.
>
> + AI-augmented Authors: A mix (i.e., >0% and <100%) of their posts were classified as AI, representing a human-AI collaborative strategy.
>
> **W4: Comparison to Prior Social Media AIGT Datasets**
>
> We have updated a comparison in the following table:
>
> | Dataset | Data Span | Timestamp | Data Source | LLM | Interaction Metadata |
> |---|---|---|---|---|---|
> | AIGTBench | 2022.1 - 2024.10 | False | Medium, Quora, Reddit | 6 | False |
> | RAID | Before 2022 | False | Wiki, News, Reddit et al. | 11 | False |
> | MultiSocial | 2018 - 2022 | False | 6 existing datasets (SMN) | 7 | False |
> | SAID | 2023.3 - 2013.10 | True | Zhihu (QA) | N/A | False |
> | RedNote-Vibe (ours) | 2020.1 - 2025.7 | True | RedNote (SMN) | 18 | Likes, Comments, Coll. |
>
> While the existing datasets are valuable, RedNote-Vibe offers unique time spans and timestamps that support AIGT evolution and trend studies. Its core novelty lies in its interaction data metadata. *Interaction count provides the weight of text on its impact.* Existing datasets treat all text with equal weight, but social media data is actually ranked, with higher-traffic posts more likely to be recommended to more people. Our dataset offers the possibility of analyzing how AIGT content can influence humans, rather than simply appearing on social media platforms.
>
> [1] http://xhslink.com/o/3ZtMUSbgoCl

---

### Official Review · Reviewer_dnne · 2025-11-01

**Soundness:** 2
**Presentation:** 2
**Contribution:** 2
**Rating:** 4
**Confidence:** 4

**Summary:**

RedNote-Vibe - the first longitudinal AIGT dataset.

**Strengths:**

The real novelty is Chinese Xiaohongshu with engagement and a 5-year span.

**Weaknesses:**

Please check the questions.

**Questions:**

Can you compare with the closest time-aware datasets? Like SAID and MultiSocial, which can do longitudinal analysis too. Only designed to longitudinal analysis is not a good novelty.

Psycholinguistic feature scores partly produced by a proxy LLM will decrease the interpretability.

Please add a human audit with inter-rater agreement, and quantify the bot risk.

Please release the exact prompts and compare with a control where the prompts vary widely.

Please add strong Chinese baselines trained under identical budgets.

Unlabeled the “exploration set”, then using PLAD to analyze. It is circular.

---

> ### Author Response · Authors · 2025-11-17
>
> To Reviewer dnne:
>
> We thank you for your time and constructive feedback on our paper. We have addressed each of your concerns below.
>
> **W1: Compare with time-aware datasets; novelty.**
>
>
> We have updated caparision in the following table:
>
> | Dataset | Data Span | Timestamp | Data Source | LLM | Interaction Metadata |
> |---|---|---|---|---|---|
> | MultiSocial | 2018 - 2022 | False | 6 existing datasets (SMN) | 7 | False |
> | SAID | 2023.3 - 2013.10 | True | Zhihu (QA) | N/A | False |
> | HC3 | N/A | False | 12 existing datasets (QA, Wiki) | 1 | False |
> | MULTITuDE | Before 2021 | False | News | 8 | -- |
> | AIGTBench | 2022.1 - 2024.10 | False | Medium, Quora, Reddit | 6 | False |
> | RAID | Before 2022 | False | Wiki, News, Reddit et al. | 11 | False |
> | RedNote-Vibe (ours) | 2020.1 - 2025.7 | True | RedNote (SMN) | 18 | Likes, Comments, Coll. |
>
>
> It can be seen that, RedNote-Vibe offers unique time spans and timestamps that supports AIGT evolution and trend studies. We emphasize its core novelty lies in its interaction data metadata. *Interaction count provides the weight of text on its impact.* Existing datasets treat all text with equal weight, but social media data is actually ranked, with higher-traffic posts more likely to be recommended to more people. Our dataset offers the possibility of analyzing how AIGT content can influence humans, rather than simply appearing on social media platforms.
>
> **W2: Proxy LLM; Human result**
>
>
> Psychometrics using LLMs is a rapidly evolving field and existing researches [1, 2] have demonstrated that modern LLMs can be more reliable and stable for psychological text analysis than traditional tools. In this work, we adopt the established practice for feature extraction with targeted prompt engeineering. Since prompt comparison has a large search space, a more meaningful comparison is the performance of different proxy models.
>
> In the following table, we report the Spearman correlation of proxy LLMs against human label on different psychological constructs.
>
> | Proxy Model | Concreteness | Sentiment | Formality |
> | --- | --- | --- | --- |
> | Qwen-turbo | 0.720 | 0.785 | 0.770 |
> | DeepSeek-V3 | 0.716 | 0.793 | 0.768 |
> | GPT-4o | 0.746 | 0.795 | 0.788 |
> | Traditional Method (LIWC-based) |	0.554 | 0.570 | 0.429 |
>
> We acknowledge that despite using the best possible methods to extract features, existing methods cannot achieve 100% accuracy. However, the PLAD framework does not rely on absolute values for interpretability; instead, it distinguishes human and AI text based on differences across various feature dimensions. For example, we found imperfection to be a key feature, but quantifying that humans are 30% less imperfect than AI seems impossible.
>
> The detailed prompts are already provided in Appendix B.2.
>
> **W3: Strong baselines.**
>
> We clarify that the baselines used in our study are strong, relevant, and appropriate for our dataset. First, fine-tuning BERT and its variants (like ROBERTa) is a standard strong baseline for AIGT detection, representing commonly-used paradigms [3, 4]. Second, we explicitly compared PLAD against the recent SOTA method LLM-Idiosyncrasies. It achieved the highest recall on the binary classification task, validating the strength of our baseline set.
>
> The significant advantage of PLAD is demonstrated in the more challenging multi-class tasks. As Table 2 clearly shows, the PLAD framework significantly outperforms all baselines on the 17-way model identification and 6-way provider identification tasks. This demonstrates PLAD is superior at capturing the subtle, intrinsic differences between models, which is a core goal of our work.
>
>
> **W4: Analyze is circular.**
>
> We respectfully disagree that this methodology is circular. It follows a standard and necessary two-step scientific process: Validation and Application.
>
> We first develop and test PLAD on our Training/Validation Set. This dataset has 100% confirmed ground-truth labels. Our experiments prove that PLAD is the most accurate and generalizable detector compared to all baselines on this known, labeled data.
>
> Then, we apply it to the real-world Exploration Set. It is the application of the best available method to analyze an unknown dataset, revealing the evolutionary trends of AI in social media.
>
> We would like to emphasize that the Exploration Set is inherently unannotable because whether a post was published using AI is unknown to anyone except the original author. However, we still believe that analyzing this data is important. This provides a baseline and data source for future AI-related social science research.
>
> [1] Large language models can infer psychological dispositions of social media users
>
> [2] JPersonaLLM: Investigating the ability of large language models to express personality traits
>
> [3] AI-generated text detection and classification based on BERT deep learning algorithm
>
> [4] Implementing bert and fine-tuned roberta to detect ai generated news by chatgpt

---

### Note · Authors · 2025-12-20

I have read and agree with the venue's withdrawal policy on behalf of myself and my co-authors.